# Ectopic Expression of *PvHMA2.1* Enhances Cadmium Tolerance in *Arabidopsis thaliana*

**DOI:** 10.3390/ijms24043544

**Published:** 2023-02-10

**Authors:** Hui Zang, Junyi He, Qi Zhang, Xue Li, Tingting Wang, Xiaojing Bi, Yunwei Zhang

**Affiliations:** College of Grassland Science and Technology, China Agricultural University, Beijing 100193, China

**Keywords:** switchgrass, PvHMA2.1, transporter, Cd tolerance, accumulation

## Abstract

Cadmium (Cd) in soil inhibits plant growth and development and even harms human health through food chain transmission. Switchgrass (*Panicum virgatum* L.), a perennial C4 biofuel crop, is considered an ideal plant for phytoremediation due to its high efficiency in removing Cd and other heavy metals from contaminated soil. The key to understanding the mechanisms of switchgrass Cd tolerance is to identify the genes involved in Cd transport. Heavy-metal ATPases (HMAs) play pivotal roles in heavy metal transport, including Cd, in *Arabidopsis thaliana* and *Oryza sativa*, but little is known about the functions of their orthologs in switchgrass. Therefore, we identified 22 *HMAs* in switchgrass, which were distributed on 12 chromosomes and divided into 4 groups using a phylogenetic analysis. Then, we focused on PvHMA2.1, which is one of the orthologs of the rice Cd transporter OsHMA2. We found that *PvHMA2.1* was widely expressed in roots, internodes, leaves, spikelets, and inflorescences, and was significantly induced in the shoots of switchgrass under Cd treatment. Moreover, PvHMA2.1 was found to have seven transmembrane domains and localized at the cell plasma membrane, indicating that it is a potential transporter. The ectopic expression of *PvHMA2.1* alleviated the reduction in primary root length and the loss of fresh weight of *Arabidopsis* seedlings under Cd treatment, suggesting that PvHMA2.1 enhanced Cd tolerance in *Arabidopsis*. The higher levels of relative water content and chlorophyll content of the transgenic lines under Cd treatment reflected that PvHMA2.1 maintained water retention capacity and alleviated photosynthesis inhibition under Cd stress in *Arabidopsis*. The roots of the *PvHMA2.1* ectopically expressed lines accumulated less Cd compared to the WT, while no significant differences were found in the Cd contents of the shoots between the transgenic lines and the WT under Cd treatment, suggesting that PvHMA2.1 reduced Cd absorption from the environment through the roots in *Arabidopsis*. Taken together, our results showed that PvHMA2.1 enhanced Cd tolerance in *Arabidopsis*, providing a promising target that could be engineered in switchgrass to repair Cd-contaminated soil.

## 1. Introduction

Cadmium (Cd) pollution in soil severely inhibits plant growth and development. Cd is absorbed via plant root cells and transferred to stems and leaves, which causes chlorophyll decomposition and leaf yellowing, resulting in plant growth restriction and even death. Cd-contaminated plants go into the food chain and are consumed by herbivorous animals and even human, leading to serious health problems [1,2,3]. Many transporter families participate in the process of Cd uptake and translocation, such as natural resistance-associated macrophage proteins (NRAMPs), ZRT/IRT-like proteins (ZIPs), ATP-binding cassette (ABC) transporters, and heavy-metal ATPase (HMA) families [4,5,6,7]. Analyses of transporter functions in regulating Cd uptake and translocation are important to uncover the molecular mechanisms of plants’ responses to Cd stress.

HMAs are members of the P_1B_-type ATPase subfamilies, which function as transporters to hydrolyze ATP for the transport of metal ions [8]. Various HMAs are characterized in different species, e.g., eight members in *Arabidopsis thaliana* named as AtHMA1-AtHMA8 [9] and nine members in *Oryza sativa* named as OsHMA1-OsHMA9 [10]. HMAs are classified into two subgroups based on their metal-substrate specificity: the zinc/cadmium/cobalt/lead group and the copper/silver group [11]. *Arabidopsis* HMA members are functionally characterized as participating in heavy metal ion transport and translocation. AtHMA1-AtHMA4 are classified into the zinc/cadmium/cobalt/lead group, while AtHMA5-AtHMA8 are categorized into the copper/silver group [12]. AtHMA1 is localized in chloroplasts and contributes to the detoxification of surplus zinc [13]. AtHMA3 is a tonoplast-localized protein involved in transferring Cd into vacuoles, which effectively prevents Cd translocation from roots to stems and grains [14]. AtHMA3 is also involved in transporting other metal ions, such as zinc, lead, and cobalt [15]. AtHMA2 and AtHMA4 are plasma membrane-localized proteins and act in root-to-shoot cadmium and zinc translocation [16,17,18]. AtHMA5 contributes to the efflux of copper from cells [19,20], while AtHMA6 (PAA1) and AtHMA7 (RAN1) are characterized as copper transporters that participate in copper translocation [21,22]. OsHMA1-OsHMA3 in rice belong to the zinc/cadmium/cobalt/lead subgroup [11], and OsHMA2 and OsHMA3 play important roles in the response to cadmium and zinc stress. OsHMA2 is localized at the plasma membrane and responsible for the transport of cadmium and zinc from roots to shoots [23]. Similar to *Arabidopsis AtHMA3*, overexpression of *OsHMA3* in rice enhances Cd tolerance, which is mainly attributed to the sequestration of Cd into root vacuoles, thus preventing Cd from moving to the shoots [24]. Overexpression of *OsHMA3* also promotes the expression of *ZIP* genes, which are involved in zinc transport, and increases the absorption of zinc [25,26]. OsHMA4-OsHMA9 belong to the copper/silver subgroup, and OsHMA4 is a tonoplast-localized protein, which can transfer copper into the vacuoles of root cells and reduce copper accumulation in rice grains [27]. *Arabidopsis* AtHMA5 and rice OsHMA5 both promote the root-to-shoot copper translocation and improve growth and development [20,28]. OsHMA6 and OsHMA9 are plasma membrane-localized proteins functioning in copper efflux, which enhances copper tolerance in rice [10,29]. 

HMAs play important roles in the response to Cd stress in distinct species, but the function of switchgrass (*Panicum virgatum* L.) HMAs in the plant’s response to Cd stress is less clear at present. Switchgrass is an important perennial C4 bioenergy grass with a high biomass, strong resistance, wide adaptability [30], and high efficiency in removing Cd from contaminated soil [31], which could be used as a promising resource for phytoremediation, i.e., the removals of pollutants from contaminated soil by plants [32]. Growing switchgrass on Cd-contaminated soil not only repairs the soil and improves ecological environment, but it also creates economic profit by providing a bioenergy resource. Previous studies on switchgrass under Cd stress mainly focused on the effects on biomass yield and physiological parameter changes [33,34]. The expression of *HMA3* was upregulated in switchgrass inoculated with plant-growth-promoting bacteria (PGPB) under Cd stress [35], while little is known about the functions and mechanisms of switchgrass HMAs in response to Cd stress. Identification of switchgrass *HMA* genes involved in the plant’s response to Cd stress is significant as it contributes to understanding the mechanisms of Cd transport and provides a potential way for the remediation of Cd-contaminated soil.

In this study, we identified 22 HMAs from switchgrass and conducted a phylogenetic analysis with 8 *Arabidopsis* HMAs and 9 rice HMAs. The chromosomal location, conserved domains, and motifs of the 22 PvHMAs were analyzed using bioinformation tools. We further identified Pavir.4KG375800.1 (PvHMA2.1) and Pavir.4NG313000.1 (PvHMA2.2) as the orthologs of rice OsHMA2, which amino acid sequences, transmembrane domains, and gene expression patterns in response to Cd treatment were highly similar. In this study, we found that PvHMA2.1 is a plasma membrane-localized protein. The ectopic expression of *PvHMA2.1* mitigated the inhibitions in primary root length, fresh weight of both shoot and root, and chlorophyll content, and maintained a high level of relative water content in *Arabidopsis* under Cd stress. Lower Cd contents in the roots of the transgenic lines indicated that the ectopic expression of *PvHMA2.1* reduced Cd absorption by roots and might promote root-to-shoot Cd translocation, leading to an alleviation of Cd toxicity in *Arabidopsis*. This study is the first to identify the function of PvHMA2.1 in *Arabidopsis*, highlighting a novel regulation mechanism of switchgrass HMAs in response to Cd stress.

## 2. Results

### 2.1. Phylogenetic Analysis of HMA Family

To identify the HMA members in switchgrass, the protein sequences of eight *Arabidopsis* HMAs (AtHMA1-AtHMA8) were retrieved from TAIR. Using a local BLASTP process, we identified 25 potential HMA members in switchgrass. Three of the HMA members were excluded because they are short in length, being only about a quarter or one-fifth of the other potential PvHMAs. To reveal the evolutionary relationship of the HMA family of switchgrass, *Arabidopsis*, and rice, the protein sequences of an additional nine rice HMAs (OsHMA1-OsHMA9) were retrieved from the Rice Genome Annotation Project. A neighbor-joining tree was conducted using the MEGA 11, based on the 39 protein sequences aligned using the online MAFFT. All HMA proteins were clustered into four groups (Group I, II, III, and IV) (Figure 1). AtHMA1 and its orthologs were grouped into Group I, including AtHMA1, OsHMA1, Pavir.4KG392400.2, and Pavir.4NG281800.2. Group II was comprised of AtHMA2, AtHMA3, AtHMA4, two orthologs from rice, and six orthologs from switchgrass. Of these proteins, four switchgrass proteins were the orthologs of OsHMA3, including Pavir.2KG146716.1, Pavir.2NG101700.1, Pavir.2KG159000.1, and Pavir.2NG101706.1, while Pavir.4KG375800.1 and Pavir.4NG313000.1 were close to OsHMA2. We named Pavir.4KG375800.1 and Pavir.4NG313000.1 as PvHMA2.1 and PvHMA2.2, which were further investigated in this study. Moreover, AtHMA6 and its orthologs (OsHMA7, Pavir.6NG288025.1, and Pavir.6KG315500.1), together with AtHMA8 and its orthologs (OsHMA8, Pavir.9NG720000.1, and Pavir.9KG615000.1) were clustered into Group III. The rest of the proteins constituted Group IV, including AtHMA5, AtHMA7, OsHMA4, OsHMA5, OsHMA6, OsHMA9, Pavir.1NG066800.1, Pavir.1KG074300.1, Pavir.1NG089700.3, Pavir.1KG102777.1, Pavir.4NG274400.1, Pavir.4KG357400.1, Pavir.7NG278500.2, Pavir.7KG284700.1, Pavir.7NG278400.1, and Pavir.7KG284600.1. The HMA members in switchgrass were at least double in number compared to those in *Arabidopsis* and rice across the four groups, which was consistent with allotetraploidy of switchgrass and diploidy of *Arabidopsis* and rice.

### 2.2. Chromosomal Location and Protein Structure of HMAs in P. virgatum

The 22 switchgrass *HMAs* are distributed on 12 chromosomes, and 1-3 of the *HMA* genes are unevenly located on each chromosome (Figure 2A). K- and N-genomes are two sets of subgenomes of switchgrass (2n = 4x = 36). Half of the *PvHMAs* are located on ChrK, while the remaining members are located on ChrN. More specifically, *Pavir.4KG392400*, *Pavir.4KG375800* (*PvHMA2.1*), and *Pavir.4KG357400* are located on Chr04K, while *Pavir.4NG281800*, *Pavir.4NG313000* (*PvHMA2.2*), and *Pavir.4NG274400* are located on the corresponding parts of Chr04N. Each two genes are present on chromosome Chr01 (*Pavir.1KG074300* and *Pavir.1KG102777* are located on Chr01K, while *Pavir.1NG066800* and *Pavir.1NG089700* are located on Chr01N), Chr02 (*Pavir.2KG146716* and *Pavir.2KG159000* are located on Chr02K, while *Pavir.2NG101700* and *Pavir.2NG101706* are located on Chr02N), and Chr07 (*Pavir.7KG284700* and *Pavir.7KG284600* are located on Chr07K, while *Pavir.7NG278500* and *Pavir.7NG278400* are located on in Chr07N), and only one gene is present on Chr06 (*Pavir.6KG315500* is located on Chr06K, while *Pavir.6NG288025* is located on Chr06N) and Chr09 (*Pavir.9KG615000* is located on Chr09K, while *Pavir.9NG720000* is located on Chr09N). The *HMA* genes exist in pair on ChrK and ChrN in terms of location, together with their evolutionary relationship (Figure 1), indicating that these paralogs might have similar functions.

The phylogenetic analysis showed a similar classification of the 22 PvHMA proteins compared to the general classification in *Arabidopsis* and rice (Figure 2B). The conserved motifs and domains of the 22 PvHMA proteins were analyzed using MEME and the Conserved Domain Search from NCBI, respectively. The numbers and varieties of motifs and domains vary across distinct groups, but they are similar within each group (Figure 2B and Appendix A). These results are helpful in the prediction of the functions of different PvHMA proteins. Rice OsHMA2 was reported to have a vital function in Cd tolerance, which was related to the root-to-shoot translocation of Cd [36]. To further investigate the functions of the OsHMA2 orthologs in switchgrass, *Pavir.4KG375800* (*PvHMA2.1*) and *Pavir.4NG313000* (*PvHMA2.2*) were cloned to study their responses to Cd stress.

### 2.3. Protein Structures and Expression Patterns of PvHMA2.1 and PvHMA2.2

To understand the potential relationships between PvHMA2.1 and PvHMA2.2, we first compared their protein sequences. The coding sequence (CDS) lengths of *PvHMA2.1* and *PvHMA2.2* are 3306 bp and 3309 bp, respectively, and each of them encodes 1101 and 1102 amino acid residues, respectively, with a similarity of 92.70% (Appendix A). Next, the transmembrane domains of PvHMA2.1 and PvHMA2.2 were analyzed using TMHMM 2.0. The transmembrane domain prediction indicates that PvHMA2.1 and PvHMA2.2 have similar transmembrane domains with a total of seven transmembrane helices (Figure 3A,B). Therefore, we speculated that PvHMA2.1 and PvHMA2.2 are potential transporters and might be functionally redundant based on their high similarities.

We further checked the tissue-specific expression patterns of *PvHMA2.1* and *PvHMA2.2* in *P. virgatum* Alamo. *PvHMA2.1* is expressed in various tissues, such as roots, internodes, leaves, spikelets, and inflorescences (Figure 3C). The expression pattern of *PvHMA2.2* shows that the highest transcript abundances are in internodes and leaves, followed by inflorescences, spikelets, and roots (Figure 3C). To test whether *PvHMA2.1* and *PvHMA2.2* respond to Cd treatment, we detected the alterations in their expression levels in switchgrass seedlings under Cd treatment. When exposed to a 7-day time-course (0, 1, 3, 5, and 7 days) treatment of 50 µM of Cd, the transcript of *PvHMA2.1* obviously increased in shoots, and the significant increase happened quickly after a 1-day treatment and reached a peak level after a 3-day treatment (Figure 3D). However, the transcript of *PvHMA2.1* decreased in roots, which changed in the opposite direction when compared to its alteration in shoots (Figure 3D). The expression of *PvHMA2.2* was significantly upregulated in both the shoots and roots of the seedlings under Cd treatment, and the peak expression level appeared after 3-day Cd treatment (Figure 3E). Although the behaviors of *PvHMA2.1* and *PvHMA2.2* in response to Cd were different in roots, they shared a quite similar trend of induction in shoots. Thus, combining the similar tissue-specific expression patterns of these two genes, we speculated that PvHMA2.1 and PvHMA2.2 play redundant roles in the response to Cd, at least in the shoots of switchgrass. Therefore, we chose PvHMA2.1 to investigate its potential function in the response to Cd.

### 2.4. Subcellular Localization of PvHMA2.1

In order to determine the subcellular localization of PvHMA2.1, the *35S:PvHMA2.1-GFP* construct or the control construct of *35S:GFP* were transiently expressed in the leaves of *Nicotiana benthamiana* to detect fluorescence using a confocal microscope. The green fluorescence of the control of GFP protein was observed not only at the cell plasma membrane but also in the nucleus (Figure 4A,B and Appendix A). Compared to the control, the green fluorescence of PvHMA2.1-GFP fusion protein was observed at the cell plasma membrane but not in the nucleus (Figure 4E,F and Appendix A). These results indicated that PvHMA2.1 was localized at the cell plasma membrane. In addition, some spots of signals in cytoplasm could be found when *35S:PvHMA2.1-GFP* was expressed (Figure 4E,F and Appendix A).

### 2.5. Ectopic Expression of PvHMA2.1 Enhanced Cadmium Tolerance in A. thaliana

In order to investigate the function of PvHMA2.1, a vector expressing *PvHMA2.1* was constructed (Appendix A). The CDS of *PvHMA2.1* was cloned from switchgrass leaves using PCR amplification (Appendix A), and digestion and ligation reactions were used for combining the *ZmUBI* promoter with *PvHMA2.1* (Appendix A). *Agrobacterium tumefaciens* GV3101 harboring *ZmUBI:PvHMA2.1* was introduced to wild-type *Arabidopsis* using the floral-dipping method (Appendix A). Seven positive transgenic lines, OM (overexpression of *PvHMA2.1*)-2, -3, -5, -6, -10, -13, and -19, were obtained from the T_1_-generation plants of *PvHMA2.1* ectopically expressed *Arabidopsis* (Appendix A). In the T_3_ generation, five transgenic lines of *Arabidopsis* displayed higher expression levels of *PvHMA2.1* compared to the WT (Appendix A).

Three independent lines (OM5-2, OM10-16, and OM19-13) were chosen to study PvHMA2.1 effect under Cd treatment (Figure 5A). Specifically, 3-day-old *Arabidopsis* seedlings of the WT and the T_4_ *PvHMA2.1* ectopically expressed lines were transferred to a 1/2 MS medium without Cd (as the control) or with 50 µM of Cd for phenotyping. After 8-day cultivation, there was no obvious difference in seedling growth between the WT and the *PvHMA2.1* ectopically expressed lines under the control condition. Under 50 µM of Cd treatment, the root growth of the WT was extremely inhibited, but that of the *PvHMA2.1* ectopically expressed lines was less restricted (Figure 5B). Moreover, the leaves of the WT were yellower than those of the *PvHMA2.1* transgenic lines under the Cd treatment (Figure 5B). To further quantify the differences, we measured the primary root lengths of *Arabidopsis* seedlings. There were no significant differences in the primary root lengths between the WT and the *PvHMA2.1* ectopically expressed lines without Cd, while the primary root lengths of the OM5-2, OM10-16, and OM19-13 lines were 126.19%, 43.86%, and 125.65% longer than those of the WT under the 50 µM Cd treatment, respectively (Figure 5C). These results indicated that PvHMA2.1 enhanced Cd tolerance in *Arabidopsis*.

### 2.6. Cadmium Contents Reduced in PvHMA2.1 Ectopically Expressed Lines in A. thaliana Roots

To test the changes in the biomass of *Arabidopsis* seedlings under Cd treatment, we measured the fresh weights of seedling roots and shoots. There were no significant differences in the fresh weights of the roots or the shoots between the WT and the *PvHMA2.1* ectopically expressed lines under the control condition. Under the Cd treatment, the fresh weights of both the roots and the shoots of three *PvHMA2.1* ectopically expressed lines were higher than those of the WT, especially the fresh weights of the OM5-2 and OM19-13 lines were significantly higher compared to the WT (Figure 6A,B). These results showed that ectopic expression of *PvHMA2.1* alleviated the loss of fresh weights in *Arabidopsis* seedlings under the Cd treatment. 

To further uncover the underlying reasons for the enhanced Cd tolerance due to the ectopic expression of *PvHMA2.1* in *Arabidopsis*, we also tested the Cd contents in the roots and shoots of the WT and the transgenic lines using the same treatments as the ones previously performed. Under the control condition, the Cd contents in the roots or shoots of the transgenic lines and the WT were similar and were at extremely low levels (Figure 6C). However, there were no significant differences in the Cd contents in the shoots between the transgenic lines and the WT, while the Cd contents in the roots of the transgenic lines were lower compared to the WT under Cd treatment (Figure 6D). The Cd contents in the roots of the OM5-2 and OM19-13 transgenic lines were significantly lower than those of the WT, and the quantitative analysis showed that the Cd contents in the roots of the OM5-2 and OM19-13 transgenic lines were 43.73% and 34.94% lower than those of the WT under the Cd treatment, respectively (Figure 6D). These results showed that the ectopic expression of *PvHMA2.1* alleviated the loss of weights and reduced Cd contents in the roots of *Arabidopsis* seedlings, which significantly enhanced Cd tolerance in *Arabidopsis*. 

### 2.7. Ectopic Expression of PvHMA2.1 Enhanced Cadmium Tolerance at the Physiological Level in A. thaliana

Physiological alteration usually reflects the damage degree of plant cells under abiotic stresses. To further study the changes at the physiological level of *Arabidopsis* seedlings under Cd treatment, the relative conductivity, malondialdehyde content, proline content, relative water content, and chlorophyll content were tested under Cd-free or Cd treatment, respectively. Relative conductivity and malondialdehyde content reflect the damage degree of cell membrane under stresses. In our study, the relative conductivity did not show any obvious difference whether with or without Cd (Figure 7A). The malondialdehyde contents in three transgenic lines were higher compared to the WT under the normal condition, but there were no differences in the malondialdehyde contents between the WT and the transgenic lines under the Cd treatment (Figure 7B). These results indicated that cell membrane integrity was not the reason for the improved Cd tolerance in the *PvHMA2.1* ectopically expressed lines during Cd exposure. Proline accumulation is another key factor that protects plants from stress-induced toxic oxygen. Our results showed that the proline content was higher in the OM10-16 line and lower in the OM19-13 line compared to the WT under the normal condition, while there were no differences in the proline contents among all seedlings under the Cd treatment (Figure 7C). These results indicated that proline accumulation was not the main reason for the improved Cd tolerance in the transgenic lines under the Cd treatment. 

In this study, the WT and the three transgenic seedlings (OM5-2, OM10-16, and OM19-13) had similar relative water contents under the control condition, while the relative water contents of the three transgenic lines were significantly higher than those of the WT during the exposure to Cd (Figure 7D), indicating that transgenic lines had a better capacity for water retention under the Cd treatment. Cd stress causes chlorophyll decomposition and leaf yellowing. The chlorophyll contents of the WT and the transgenic lines were similar in the absence of Cd, while the chlorophyll contents of the WT were lower than those of the transgenic lines under the Cd treatment (Figure 7E), indicating that the transgenic lines had a stronger photosynthetic system under the Cd treatment. These results showed that the higher level of relative water content and the lower reduction in chlorophyll content were related to an enhancement in Cd tolerance in the transgenic lines under the Cd treatment. Therefore, we demonstrated that the ectopic expression of *PvHMA2.1* also enhanced Cd tolerance at the physiological level in *Arabidopsis*, especially in terms of maintaining the high level of relative water content and alleviating the reduction in chlorophyll content. Taken together, the ectopic expression of *PvHMA2.1* gene significantly enhanced Cd resistance in *Arabidopsis*.

## 3. Discussion

Cd is an unessential trace element for plants, which is inadvertently taken up through the absorption pathways of some essential divalent metal ions and affects many aspects of plant physiology and biochemistry [37]. The detoxification mechanism of Cd is crucial for the normal growth of plants. The heavy-metal ATPases (HMAs), P_1B_-type ATPase members, have been proven to play pivotal roles in Cd detoxification in many plants [26,38,39]. However, the functions of switchgrass HMAs have not been identified yet. In this study, we characterized one of the switchgrass HMAs, named PvHMA2.1, and unveiled its critical role in enhancing Cd tolerance in *Arabidopsis*.

### 3.1. Analyses of Classification and Protein Structures of P. virgatum HMAs

The 39 HMA proteins (8, 9, and 22 HMA members from *Arabidopsis*, rice, and switchgrass, respectively) were classified into four groups according to the phylogenetic analysis (Figure 1), which is consistent with previous reports on *Arabidopsis* and rice [9,10], indicating that the structures of HMA proteins are conserved in switchgrass, *Arabidopsis*, and rice. Various HMAs in *Arabidopsis* and rice play different roles under heavy metal stress [11]. AtHMA2 maintains zinc homeostasis by promoting the efflux of excess zinc [40], and AtHMA2 is also activated by Cd and interacts with Cd [41]. AtHMA3 enhances Cd tolerance by isolating excess Cd into vacuoles [42], while AtHMA4 confers Cd resistance in yeast and improves tolerance to cadmium and excess zinc by binding metal ions in *Arabidopsis* [43,44]. In rice, OsHMA2 plays an important role in transporting cadmium and zinc from roots to shoots [23]. Overexpression of *OsHMA3* leads to Cd isolation into vacuoles and a reduction in Cd accumulation in grains, thus improving Cd tolerance in rice [26]. Switchgrass HMA proteins have closer relationships to rice HMA proteins than those of *Arabidopsis* within each group (Figure 1), indicating that the evolutionary relationship of HMA family members among monocotyledons is closer. The structure analysis of the switchgrass HMA proteins shows that the numbers and varieties of motifs and domains are highly similar within each group (Figure 2B and Appendix A), indicating that the structures of switchgrass HMA proteins are conserved in each group. Therefore, we speculate that the functions of switchgrass HMA proteins and their orthologs in rice may be conserved. The HMA proteins respond to different heavy metal stresses in rice [12], and the ones responding to Cd stress mainly include OsHMA2 and OsHMA3 [45,46].

As previously described, the known members of HMA proteins in Group II, including AtHMA2, AtHMA3, AtHMA4, OsHMA2, and OsHMA3, are more relevant than other group members in response to Cd stress [11,24]. In order to investigate the effects of Cd stress on switchgrass, our study focused on the six HMA proteins of switchgrass in Group II (Figure 1). OsHMA2 plays an important role in Cd transport from roots to shoots and enhances Cd tolerance in rice [36], which orthologs in switchgrass might have similar functions. Therefore, we identified two HMA proteins of switchgrass in Group II, Pavir.4KG375800.1 (PvHMA2.1) and Pavir.4NG313000.1 (PvHMA2.2), which are orthologous to OsHMA2 in rice (Figure 1). Due to the high similarities in amino acid sequences (Appendix A) and the gene expression patterns of PvHMA2.1 and PvHMA2.2 under Cd treatment (Figure 3D,E), we chose to further study PvHMA2.1, which might be functionally redundant with PvHMA2.2. Similar to the subcellular localization of OsHMA2, PvHMA2.1 is also localized at the cell plasma membrane (Figure 4 and Appendix A), which is consistent with the speculation that PvHMA2.1 is a transmembrane protein (Figure 3A), providing further proofs for PvHMA2.1 as a transporter. Therefore, we assume that PvHMA2.1 plays a vital role in Cd tolerance as a transmembrane protein. In addition, we found that the green fluorescence of PvHMA2.1-GFP fusion protein also appeared as spots in the cytoplasm (Figure 4E,F), which indicated that PvHMA2.1 might be localized in chloroplast as well. In *Sedum plumbizincicola*, SpHMA1 is localized at chloroplast envelope to alleviate Cd toxicity there [47]. Therefore, based on the findings regarding homology and subcellular localization, we speculate that PvHMA2.1 plays an important role, similar to OsHMA2 and SpHMA1, in Cd tolerance, but this needs further verification.

### 3.2. Ectopic Expression of PvHMA2.1 Enhances Cadmium Tolerance in A. thaliana

In order to reveal the function of PvHMA2.1, the constructs overexpressing *PvHMA2.1* were transformed into *Arabidopsis*. The root lengths and the fresh weights of the *PvHMA2.1* ectopically expressed *Arabidopsis* seedlings were less inhibited than those of the WT under the Cd treatment (Figure 5C and Figure 6A,B), indicating that PvHMA2.1 alleviates the inhibitions in root and shoot growth by Cd stress in *Arabidopsis*. Therefore, PvHMA2.1 enhances Cd tolerance in *Arabidopsis*. By testing the internal Cd content, we found that the Cd contents in the roots of the *PvHMA2.1* ectopically expressed lines were lower than those of the WT, while the Cd contents in the shoots of the transgenic lines and the WT were similar under the Cd treatment (Figure 6D), suggesting that PvHMA2.1 reduces Cd absorption in roots and may simultaneously promote Cd transport from roots to shoots in *Arabidopsis*, similar to the function of OsHMA2 in root-to-shoot Cd translocation [45].

Cd toxicity is harmful to plants by disturbing the overall physiological mechanism. Relative conductivity indicates the damage degree of plant cells under stresses [48]. Malondialdehyde content is an indicator of lipid peroxidation, which reflects various external stimuli [49]. Proline is an important osmotic protective substance that plays a key role in maintaining the stability of biological structures and cell functions [50]. Our results showed that there were no differences in relative conductivity, malondialdehyde content, and proline content between the transgenic lines and the WT under the Cd treatment (Figure 7A–C), indicating that these physiological parameters are not the reason for the enhanced Cd tolerance in the transgenic lines under the Cd treatment. The relationship between proline accumulation and abiotic stress tolerance is still controversial. In this study, the proline contents of all *PvHMA2.1* ectopically expressed lines did not increase under the Cd treatment (Figure 7C), which indicates that *Arabidopsis* seedlings under Cd stress might improve the ability to maintain proline homeostasis [51]. Relative water content indicates the water holding capacity of plant cells under stresses and also reflects the stress degree [52]. The relative water contents of the *PvHMA2.1* transgenic lines were significantly higher than those of the WT under Cd stress (Figure 7D), indicating that PvHMA2.1 enhances water holding capacity and Cd tolerance under Cd stress. Photosynthesis is a major process for plant growth and development, which is sensitive to heavy metal stress. Excessive Cd accumulation in plants can cause membrane damage in chloroplast [53]. The chlorophyll contents in all seedlings decreased under the Cd treatment (Figure 7E), showing that Cd toxicity destroys chloroplast and further affects plant photosynthesis, while the chlorophyll contents of the transgenic lines were higher than those of the WT under the Cd treatment (Figure 7E), indicating that the ectopic expression of *PvHMA2.1* alleviates the damages of Cd stress on chloroplast and enhances Cd tolerance in the transgenic lines. SpHMA1 is localized at chloroplast envelope and functions to protect photosynthesis by preventing Cd accumulation in chloroplast [47]. The results of subcellular localization indicated that PvHMA2.1 might be also localized in chloroplast (Figure 4E,F), so we speculate that PvHMA2.1 plays an important role in the photosynthetic system of plants to enhance Cd tolerance.

### 3.3. Expression of PvHMA2.1 Homologous Genes Is Induced by Cadmium Stress

In this study, we only identified the function of PvHMA2.1 in *Arabidopsis*, which is homologous to OsHMA2. PvHMA2.2 is similar to PvHMA2.1 in many aspects, so we speculate that PvHMA2.2 may be functionally redundant with PvHMA2.1, but this still needs verification. The HMA members of *Arabidopsis* and rice in Group II are all relevant in the response to Cd stress, so we wanted to know whether the other four HMAs of switchgrass in Group II also respond to Cd stress. We further analyzed the four HMA proteins of switchgrass in Group II, including Pavir.2NG101706.1, Pavir.2KG159000.1, Pavir.2NG101700.1, and Pavir.2KG146716.1, which are orthologous to rice OsHMA3 (Figure 1). When exposed to a 7-day time-course treatment of 50 µM of Cd, the expression levels of *Pavir.2NG101706*, *Pavir.2KG159000*, *Pavir.2NG101700*, and *Pavir.2KG146716* in switchgrass seedlings were significantly upregulated in the shoots or roots (Appendix A). These results indicated that the four *HMA* genes of switchgrass could also respond to Cd stress. It is speculated that their functions may be similar to rice OsHMA3, which reduces root-to-shoot translocation of Cd and enhances Cd tolerance [54]. However, the speculated functions of the four PvHMA proteins still need to be verified in future research.

## 4. Materials and Methods

### 4.1. Analyses of Gene and Protein Sequences of HMAs

The protein sequences of eight *Arabidopsis* HMAs (AtHMA1-AtHMA8) were downloaded from TAIR (https://www.arabidopsis.org/, accessed on 31 October 2022). Using them as the queries, a local BLASTP against the annotated protein sequences of switchgrass genome of *P. virgatum* v5.1 from phytozome 13 (https://phytozome-next.jgi.doe.gov/, accessed on 31 October 2022) was conducted, with an e-value cutoff of 0.05 and identity of 50%. In total, 25 potential PvHMAs were found in switchgrass, while 3 (Pavir.9NG549714.1, Pavir.2NG595320.1, and Pavir.2KG546500.6) out of 25 candidate PvHMAs were excluded from further analyses due to their shortness in sequence length.

To further investigate the evolutionary relationship of the remaining 22 PvHMAs, multiple sequence alignment was performed with the protein sequences of the 22 switchgrass HMAs, 8 *Arabidopsis* HMAs, and 9 rice HMAs (OsHMA1-OsHMA9), which were retrieved from the Rice Genome Annotation Project (http://rice.plantbiology.msu.edu/annotation.shtml, accessed on 31 October 2022), using MAFFT (https://www.ebi.ac.uk/Tools/msa/mafft/, accessed on 9 November 2022) with default parameters, and a phylogenetic tree was constructed using the MEGA 11, using the neighbor-joining method with 1000 bootstrap replicates. The chromosomal location of the 22 *PvHMA* genes was visualized using Gene Location Visualize from the GTF/GFF module of TBtools [55]. The conserved motifs and domains of the 22 PvHMAs were analyzed using MEME (https://meme-suite.org/meme/tools/meme, accessed on 9 November 2022) with 10 motifs and the Conserved Domain Search from NCBI (https://www.ncbi.nlm.nih.gov/Structure/cdd/wrpsb.cgi, accessed on 10 November 2022) with the initial parameters, respectively. The TBtools was used to visualize the conserved motifs and domains [55]. Amino acid sequence alignment and transmembrane domain prediction of PvHMA2.1 and PvHMA2.2 were performed using DNAMAN 6 software and TMHMM 2.0 (https://services.healthtech.dtu.dk/service.php?TMHMM-2.0, accessed on 10 November 2022) with the initial parameters, respectively. The sequence numbers of HMAs in *P. virgatum*, *A. thaliana*, and *O. sativa* are listed in the Appendix A.

### 4.2. Materials and Growth Conditions of P. virgatum

*P. virgatum* cultivar Alamo was used in this study. Switchgrass wild-type plants were cultivated in a greenhouse with natural light and night-day temperature ranging from 25–35 °C. Roots, internodes, leaves, and spikelets at reproductive stage 1 (R1), and inflorescences at R3 were harvested for RNA extraction and determination of tissue-specific expression patterns of *PvHMA2.1* and *PvHMA2.2* [56].

Surface-sterilized seeds of wild-type switchgrass were placed in the dark for 3 days at 4 °C, and then transferred to the wet filter paper to germinate and grow for 7 days. The seedlings were transferred to 1/4 Hoagland’s nutrient solution for another 7 days, and then treated under 50 µM CdCl_2_ (0 µM CdCl_2_ as the control condition) in a growth chamber under 25 °C (16 h light/8 h dark cycles). During 7-day time-course (0, 1, 3, 5, and 7 days) Cd treatment, the seedlings of switchgrass were quickly rinsed with deionized water for three times, absorbed surface moisture with filter papers, placed in centrifuge tubes, frozen in liquid nitrogen, and stored at −80 °C for RNA extraction.

### 4.3. Transcript Analysis by Real-Time Quantitative PCR

Total RNA was extracted using the Quick General Plant RNA Extraction Kit (Huayueyang, Beijing, China) and then reverse-transcribed to cDNA using the PrimeScript™ RT reagent Kit (Perfect Real Time; TaKaRa, Beijing, China). Real-time fluorescent quantitative PCR (qPCR) was performed using the TB Green™ Premix Ex Taq™ II Kit (TaKaRa, China). *PvUbiquitin* (GenBank: FL955474.1) and *AtActin2* (AT3G18780) were used as the references in switchgrass and *Arabidopsis*, respectively. The primers for qPCR are listed in Appendix A. Relative gene expression calculations were conducted using 2^−ΔΔCT^ [57].

### 4.4. Subcellular Localization of PvHMA2.1

*PvHMA2.1* CDS was amplified using PCR, which was cloned into the vector of *pMDC83-GFP* driven by the *35S* promoter, using the specific primers listed in Appendix A. The constructs of *35S:PvHMA2.1-GFP*, the control of *35S:GFP*, *p19*, the cell plasma membrane marker *CBL-RFP*, and the nucleus marker *Athook-RFP* were introduced into *agrobacterium* strain GV2260, respectively. The mixtures of *35S:PvHMA2.1-GFP*, *p19*, and *CBL-RFP*/*Athook-RFP* were transiently expressed in the leaves of 3-week-old *N. benthamiana*, and the mixtures of *35S:GFP*, *p19* and *CBL-RFP*/*Athook-RFP* were used as the controls. The subcellular localization of PvHMA2.1 was visualized using a confocal microscope after 2–3 days of infiltration.

### 4.5. Expression Vector Construction and Selection for A. thaliana

*PvHMA2.1* CDS was cloned into the plant binary vector *pUbi1301* [58] via the restriction enzymes of *Bam*HI and *Kpn*I, using the specific primers listed in Appendix A. *pUbi1301-PvHMA2.1* was transformed into *agrobacterium* strain GV3101, which was subsequently introduced to wild-type *Arabidopsis* Columbia by the floral-dipping method [59,60]. The selection of putative *Arabidopsis* transgenic plants was performed on a ½ MS medium with 50 mg/L of hygromycin and 100 mg/L of ampicillin and further confirmed using PCR and qPCR. Three independent transgenic lines with elevated expression levels of *PvHMA2.1* were used for the investigation of *PvHMA2.1* in response to Cd stress.

### 4.6. Evaluation of Cadmium Tolerance in A. thaliana

The *Arabidopsis* T_4_ transgenic lines (OM5-2, OM10-16, and OM19-13) and the WT seeds were sterilized with 6% NaClO for 10–15 min and placed in the dark for 3 days at 4 °C. The surface-sterilized seeds were grown on a 1/2 MS medium for 3 days and transferred to a 1/2 MS medium without Cd or with 50 µM of CdCl_2_ for another 8 days. The primary root lengths of all seedlings were measured individually using IMAGEJ 1.52 after the photographs were taken. The shoots and roots of the seedlings were sampled, with 10 plants in each replicate, and their fresh weights were measured after removing surface moisture by filter papers. Four samples were repeated. 

To quantify the effect of Cd toxicity, the relative conductivity and the malondialdehyde content of the *Arabidopsis* seedlings, with 10 plants in each replicate, were measured according to the methods mentioned in [61]. The proline content, relative water content, and chlorophyll content of the *Arabidopsis* seedlings were tested as described previously [52]. Four biological replicates were measured for each parameter.

### 4.7. Determination of Cadmium Contents in A. thaliana

*Arabidopsis* seedlings were grown on a 1/2 MS medium without Cd or with 50 µM of CdCl_2_ for 8 days. The shoots and roots of the *Arabidopsis* seedlings, with 10 plants in each replicate, were rinsed by deionized water three times, and the fresh weights of the shoots and roots were recorded. Next, the fresh samples were dehydrated for 6 h at 80 °C and digested in HNO_3_ [62]. Cd contents were determined using an inductively coupled plasma mass spectrometry (ICP-MS) after filtered by 0.22 μm filter membrane. Three biological samples were repeated. 

### 4.8. Statistical Analyses

Statistical analyses were performed using Excel and GraphPad Prism 9. The values are represented as mean ± SD. Significance of differences was determined using Student’s t-test, as indicated in figures. One, two, and three asterisks indicate *p* < 0.05, *p* < 0.01, and *p* < 0.001, respectively.

## 5. Conclusions

Taken together, the ectopic expression of *PvHMA2.1* in *Arabidopsis* reduces Cd absorption in roots and may promote Cd transport from roots to shoots, thereby enhancing Cd tolerance in *Arabidopsis*. At the physiological level, the enhancement of Cd tolerance in the *PvHMA2.1* ectopically expressed lines is due to the maintenance of water holding capacity and the photosynthesis stability of *Arabidopsis* seedlings by *PvHMA2.1* under Cd stress. PvHMA2.1 enhances Cd tolerance and may promote Cd transport from roots to shoots in *Arabidopsis*, while its function in switchgrass is still unknown. In future research, we will continue to investigate the function of PvHMA2.1 in switchgrass, which may be of great significance for the use of switchgrass as a resource to repair Cd-contaminated soil.

## Figures and Tables

**Figure 1 ijms-24-03544-f001:**
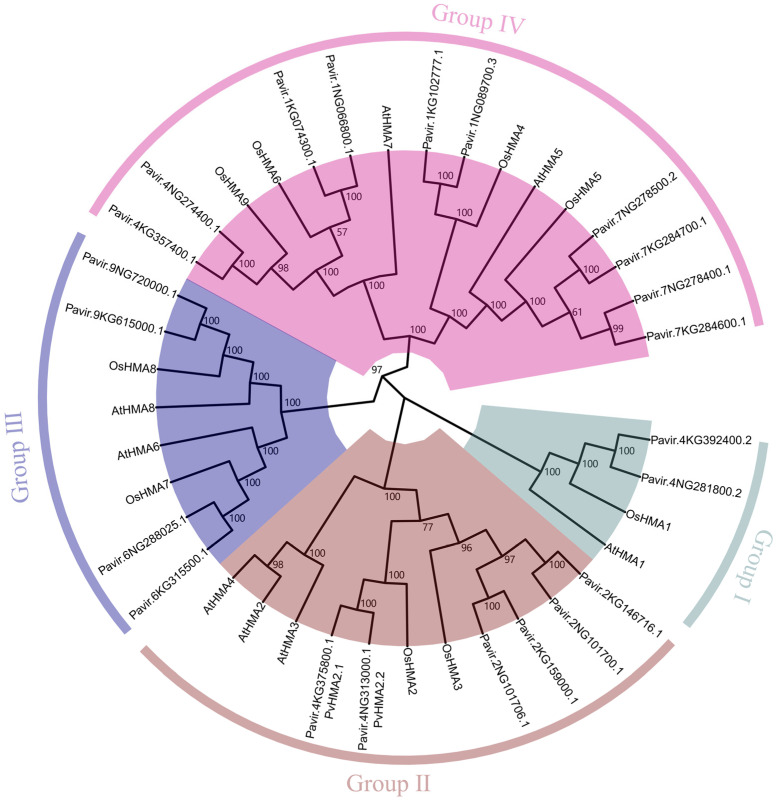
The unrooted phylogenetic tree for heavy-metal ATPase (HMA) proteins in *Panicum virgatum*, *Arabidopsis thaliana*, and *Oryza sativa*. Multiple HMA sequence alignment was first performed using MAFFT, and then a phylogenetic tree was constructed by MEGA 11, using the neighbor-joining method with 1000 bootstrap replicates. The numbers next to the branching point indicate the percentage of replicates supporting each branch. All HMAs are classified into four groups, which are marked by different colors. The sequence numbers of HMAs in *P. virgatum*, *A. thaliana*, and *O. sativa* are listed in Appendix A.

**Figure 2 ijms-24-03544-f002:**
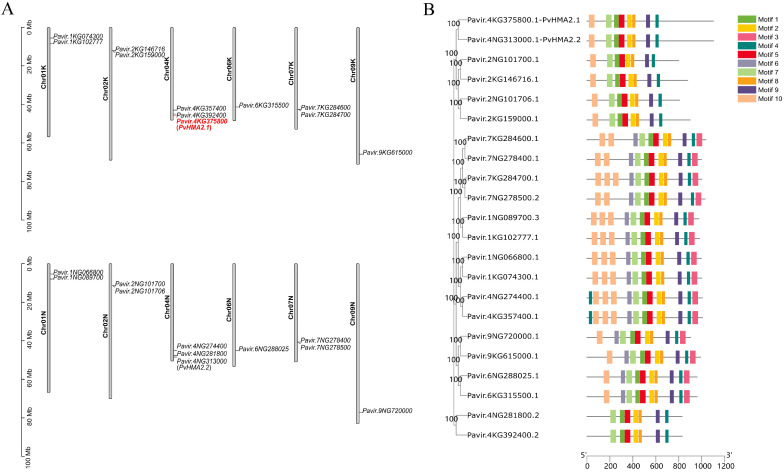
Chromosomal position and conserved motifs of 22 *HMAs* in *P. virgatum*. (**A**) Chromosomal mapping of *HMA* genes in *P. virgatum*. Chromosome name and gene identifier are indicated on the left and right of each chromosome, respectively. The ordinate is used as a reference for chromosome length. (**B**) Phylogenetic tree and conserved motifs of switchgrass HMA proteins. The tree of the HMA proteins was constructed using the multiple sequence alignment of MAFFT, and the phylogenetic tree was constructed using MEGA 11, using the neighbor-joining method with 1000 bootstrap replicates. The numbers next to the branching point indicate the percentage of replicates supporting each branch. The conserved motifs of the HMA proteins were analyzed by using the online program MEME with ten motifs. The abscissa represents the scale for sequence length of the HMA proteins, and the boxes with different colors represent different motifs in the corresponding position of each protein.

**Figure 3 ijms-24-03544-f003:**
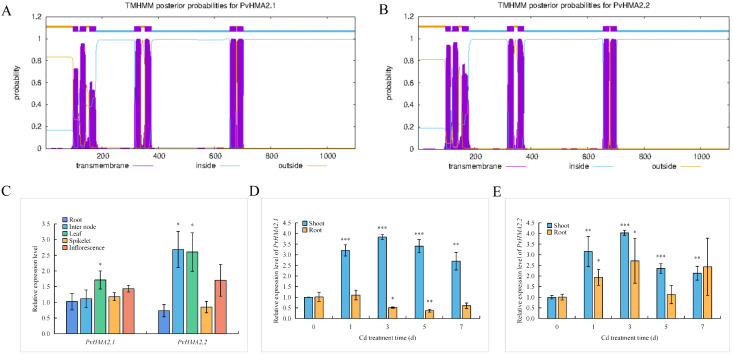
Protein transmembrane domain prediction and expression patterns of *PvHMA2.1* and *PvHMA2.2*. (**A**,**B**) Predictions of PvHMA2.1 (**A**) and PvHMA2.2 (**B**) protein transmembrane domains using TMHMM 2.0. The abscissa represents the amino acid sequence location. The ordinate represents the probability of the amino acids being located inside the membrane, outside the membrane, or in the transmembrane region. The purple parts represent the transmembrane helix structures, the blue parts represent the structures located inside the membrane, and the brown parts represent the structures located outside the membrane. (**C**) The relative expression levels of *PvHMA2.1* and *PvHMA2.2* in roots, internodes, leaves, spikelets, and inflorescences of *P. virgatum* Alamo. *PvHMA2.1* gene expression levels in roots are normalized as 1. (**D**,**E**) The relative expression levels of *PvHMA2.1* (**D**) and *PvHMA2.2* (**E**) in shoots and roots in response to a time-course cadmium (Cd) treatment. Specifically, 14-day-old switchgrass seedlings were grown in a 1/4 Hoagland’s nutrient solution without Cd (as the control) or with 50 µM of Cd for 7 days. The transcript levels of *PvHMA2.1* and *PvHMA2.2* under the control condition are normalized as 1. Switchgrass *Ubiquitin* expression is used as an internal reference, and the primers used for the qPCR are listed in Appendix A. Error bars represent the standard deviations of the three replicates. Asterisks indicate significant differences between the treatment group and the control group: one asterisk indicates *p* < 0.05, two asterisks indicate *p* < 0.01, and three asterisks indicate *p* < 0.001.

**Figure 4 ijms-24-03544-f004:**
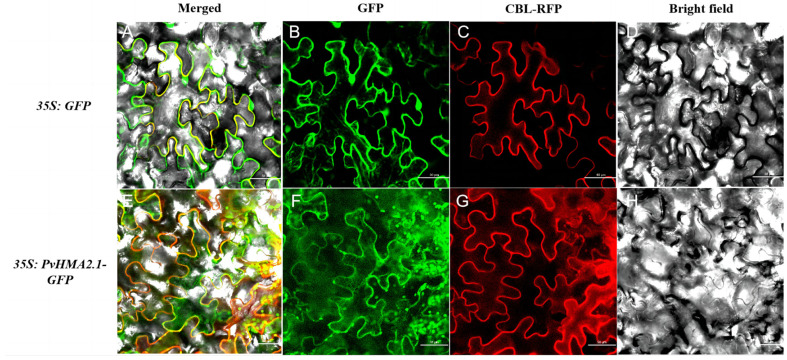
Subcellular localization of PvHMA2.1 in tobacco leaf cell. (**A**) The merger of GFP protein green fluorescence, the cell plasma membrane marker (CBL) protein red fluorescence, and the bright field. (**B**) The green fluorescence of GFP protein. (**E**) The merger of PvHMA2.1-GFP fusion protein green fluorescence, the cell plasma membrane marker (CBL) protein red fluorescence, and the bright field. (**F**) The green fluorescence of PvHMA2.1-GFP fusion protein. (**C**,**G**) The red fluorescence of cell plasma membrane marker (CBL) protein. (**D**,**H**) Bright field. Scale bars are 50 μm.

**Figure 5 ijms-24-03544-f005:**
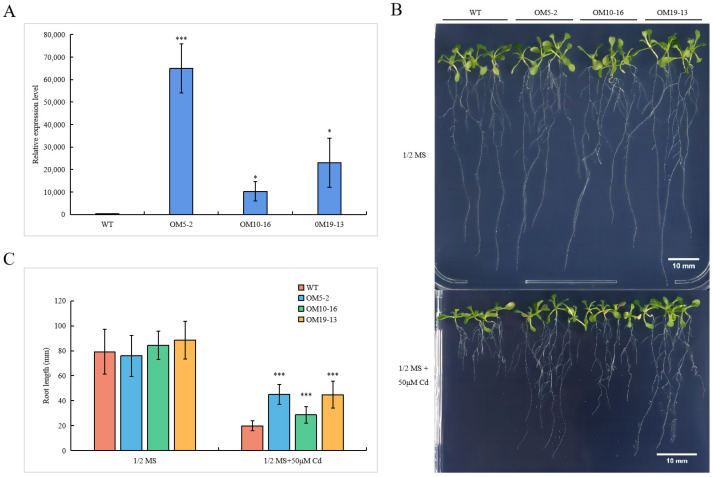
Analysis of *PvHMA2.1* ectopically expressed lines in response to Cd treatment in *A. thaliana*. (**A**) Relative expression levels of *PvHMA2.1* in the *PvHMA2.1* ectopically expressed lines and the WT in *Arabidopsis* without Cd treatment. Specifically, 7-day-old *Arabidopsis* seedlings of the T_3_ *PvHMA2.1* ectopically expressed lines and the WT grown on a 1/2 MS medium were harvested for transcription detection using qPCR. The expression levels of *PvHMA2.1* gene in the WT are normalized as 1. Error bars represent the standard deviations of the three replicates. (**B**) Phenotypic analysis of *Arabidopsis* seedlings under Cd stress. Specifically, 3-day-old *Arabidopsis* seedlings of the T_4_ transgenic lines (OM5-2, OM10-16, and OM19-13) and the WT were grown in a 1/2 MS medium without Cd or with 50 µM of Cd for 8 days, and photographs were taken. Scale bars are 10 mm. (**C**) Primary root lengths of *Arabidopsis* seedlings under Cd stress. Primary root lengths were measured using IMAGEJ 1.52, and the data represent the mean ± SD, *n* ≥ 21. Asterisks indicate significant differences: one asterisk indicates *p* < 0.05, and three asterisks indicate *p* < 0.001.

**Figure 6 ijms-24-03544-f006:**
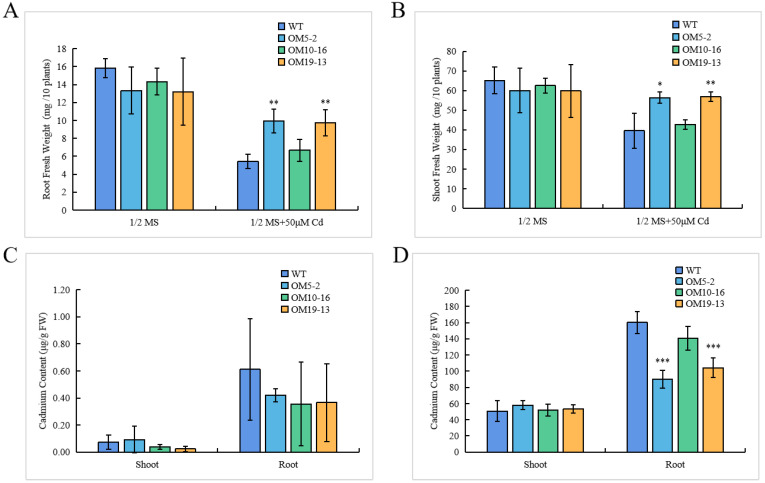
Seedlings weights and Cd contents of the WT and the *PvHMA2.1* ectopically expressed lines in *A. thaliana*. (**A**,**B**) Fresh weights of roots (**A**) and shoots (**B**) in *Arabidopsis* seedlings grown on a 1/2 MS medium without Cd (as the control) or with 50 µM of Cd for 8 days. (**C**,**D**) Cd contents in the shoots and roots of *Arabidopsis* seedlings grown on a 1/2 MS medium without Cd (**C**) and with 50 µM of Cd (**D**) for 8 days. Cd contents were determined by using an inductively coupled plasma mass spectrometry (ICP-MS). The data represent the mean ± SD, *n* ≥ 3. Asterisks indicate significant differences compared to the WT: one asterisk indicates *p* < 0.05, two asterisks indicate *p* < 0.01, and three asterisks indicate *p* < 0.001.

**Figure 7 ijms-24-03544-f007:**
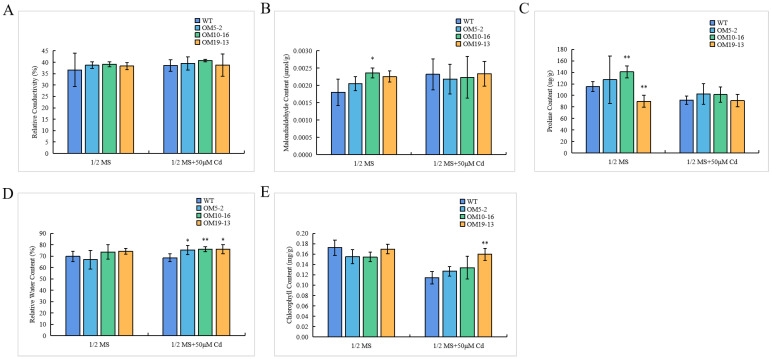
Physiological parameter analysis of *Arabidopsis* seedlings under Cd treatment. (**A**–**E**) Relative conductivity (**A**), malondialdehyde content (**B**), proline content (**C**), relative water content (**D**), and chlorophyll content (**E**) of *Arabidopsis* seedlings grown on a 1/2 MS medium without Cd (as the control) or with 50 µM of Cd for 8 days. WT, wild-type *Arabidopsis*; OM5-2, OM10-16, and OM19-13, T_4_ *PvHMA2.1* ectopically expressed *Arabidopsis* lines. The data represent the mean ± SD, *n* = 4. Asterisks indicate significant differences from the WT: one asterisk indicates *p* < 0.05, and two asterisks indicate *p* < 0.01.

## Data Availability

The data presented in this study are available in the Appendix A.

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
