# Peer review of "Ectopic Expression of PvHMA2.1 Enhances Cadmium Tolerance in Arabidopsis thaliana"

_ijms, 2023, doi:10.3390/ijms24043544_

Round 1

Reviewer 1 Report

The research article written by Zang et al addresses an important problem of Cadmium toxicity in the soil and how it can be eliminated by plants. In this paper, the authors have identified genes that code for Heavy metal ATPases (HMAs) and looked into how they play key role in the uptake of cadmium.

The author’s analyzed several HMAs by computational analysis and chose a gene to study in details. The gene has been shown to express higher in the presence of cadmium and implicated to code a transporter. They then cloned and expressed the potential switchgrass transporter, PvHMA2.1 in Arabidopsis and studied the effects of overexpression. The author’s performed localization studies for PvHMA2.1 and showed that it localizes to the plasma membrane implying that it could be a transporter.

They carried out physiological studies and observed that ectopic expression in Arabidopsis reduced Cd absorption in roots and also indicated that there was promotion of Cd transport from roots to shoots. The authors also show that overexpressed lines has increased capacity of water consumption and robust photosynthesis along with some other physiological parameters.

This was indicative of increased tolerance of these plants to cadmium.

The paper has some significant findings and must be published. It has valuable information in being able to grow plants in Cd infested soils. However, I highly recommend proof reading the article for errors in “English tense” and the manner in which sentences are written. It took me a while to read, reread and understand the sentences in the article.

For the same reason, I suggest that the Title of the paper could be revisited as well.

Reviewer 2 Report

In this study, the authors focused on heavy metal ATPase (HMA) proteins in Panicum virgatum. More specifically, the expression and function of PvHMA2.1 was explored in response to Cd treatment. To validate the function of PvHMA2.1 as a transporter allowing Cd resistance, this gene was expressed in Arabidopsis thaliana. The study was very well conducted, so I only have the following comments and suggestions.

-        Because PvHMA2.2 expression is induced while PvHMA2.1 expression is reduced in roots upon Cd treatment, I think that it would have been more interesting to study the function of PvHMA2.2.

-        Lines 83-84: ``However, there are few studies on switchgrass HMAs involved in Cd resistance at present.`` References should be provided here.

-        Figures 2 and 3: the resolution of the figures should be increased.

-        Table S2: this table should be moved into the material and method section.

-        Figure 5 A: the figure legend should indicate if a Cd treatment was done on the seedlings or not.

-        Figure S4 C: + and – should be explained in the figure legend.

-        Lines 139-140: K- and N-genomes should be defined.

-        Line 297: “when treated without Cd”. What kind of treatment is it?

-        Line 351: “ICP-MS » this abbreviation should be explained

-        The English language should be significantly improved throughout the manuscript. Examples:

o   The beginning of the title seems unclear to me and should be corrected, maybe as “Ectopic expression of Switchgrass PvHMA2.1 Enhances Cadmium Tolerance in Arabidopsis thaliana”.

o   Lines 22-24: “Ectopic expressing PvHMA2.1 in Arabidopsis reduced Cd accumulation in seedling roots but not happened so in the shoots, suggesting that PvHMA2.1 reduced Cd absorption from environment through roots in Arabidopsis.” This sentence should be revised.

o   Line 298: “the roots of WT were extremely limited” should be revised.

Reviewer 3 Report

The text require significant corrections. Below some comments.

Please, check carefully style.

Lines 7-9: it is better to separate two messages to two sentences.

Line 9: it is better to link Panicum virgatum with switchgrass: like “Switchgrass (Panicum virgatum L.)…”

Line 10: these two sentences does not directly follow each other.

Line 11: you can not systematically indenitified,  you can systematically investigated.

Line 15: “PvHMA2.1 in switchgrass” . Pv already means switchgrass. It is not necessary to write it again.

Line 18: “PvHMA2.1 increased the primary root length” – this is not try, ectopic expresión prevent root growth inhibition.

Line 23: “but not happened so in the shoots”?  Please, reformulate.

Ectopic expressión means that all cell expressed transporter. The question: does shoots directly contact Cd-contain médium? Or absorbtion occurred through root only? If so, it may means that transporter effectively translocate Cd to shoot. This led to more developed root system with more cytokinin and other essential elements move to shoots and promoting it development.

Line 24-27: please, reformulate more clear way.

Line 36:  “plant Cd uptake” – there is no “plant Cd”, please, delete Word “plant”.

Line 43: “in species”?

Line 51: vacuole membrane = tonoplast

Line 73- 84: this part did not have a clear strusture, jump form one point to another chaotically.

Lines 102 and 107: “were obtained”? It is better to write “were received”.

Line 139:  “of switchgrass” – here you described only switchgrass, so, you do not need to mention it again.

Line 181: similar comment.

Line 235: “fluorescence using confocal.” C- please, do not use lab slang. Confocal microscope. 

Line 293: “OM5-2, OM10-16, and OM19-13 three independent lines” = “Three independent lines (OM5-2, OM10-16, and OM19-13)…”

Line 294: “function under Cd treatment”? You study effect, not function.

Line 297: “when treated without Cd” - = in the control.

Line 299: “the roots of WT were extremely limited”? Root can not be limeted itself. 

Line 325: “increased” – not increased, but keep higher!

Line 338: “Under Cd treatment, Cd content of shoots between WT and transgenic lines 338 displayed no significant difference, while lower Cd content of roots in three transgenic 339 lines was shown (Figure 6F).” – please, re-formulate!

Line 522: “Using protein sequences of Arabidopsis eight HMAs (AtHMA1-AtHMA8) obtained” ??

Round 2

Reviewer 3 Report

The text is better now, thank you!

Please, clarify the following points:

Figure 3 C: the normalization method is not clear. How did you normalise for lines 2.2? It looks like in root expression level is less as 1.

It will be also nice to provide a comparison with expression level in the control.

Fig 5 B, C: I am not sure that shoot and root weight so low… Please, clarify.

Line 481: I am not sure this comparison is correct: they seen effect at 500 µM Cd, while in your case it is only 50 µM. Moreover, term „dissolve the thylakoid membrane“ is not very lear, it is rather membrane damage.

Figure 4: there are a contradiction with other figure: the prootein should localse in the tonoplast, not only on the plasma membranes. Please, clarify this point.

It will be nice to have expression level in Arabidopsis in relation to organ specificty: root vs shoot.

There are also another contradiction: on figure 3 gene expression in shoots were higher in the shoots, while in Arabidopsis Cd accumulation mainly attributed to the root. Please, clarify.

Round 3

Reviewer 3 Report

The text is better now, please, change figure 6 only.

Figure 6: it looks like 10 plants dry weight is only 0.2 mg. How accurate were measurements?

I would suggest to use mg to avoid confusion and too much „0“ in the X scale.
